# New Challenges in Heart Failure with Reduced Ejection Fraction: Managing Worsening Events

**DOI:** 10.3390/jcm12226956

**Published:** 2023-11-07

**Authors:** Carlo Lavalle, Luca Di Lullo, Jean Pierre Jabbour, Marta Palombi, Sara Trivigno, Marco Valerio Mariani, Francesco Summaria, Paolo Severino, Roberto Badagliacca, Fabio Miraldi, Antonio Bellasi, Carmine Dario Vizza

**Affiliations:** 1Department of Clinical, Internal, Anesthesiologist and Cardiovascular Sciences, Sapienza University of Rome, Viale del Policlinico 155, 00161 Rome, Italy; carlo.lavalle@uniroma1.it (C.L.); jeanpierre.jabbour@uniroma1.it (J.P.J.); marta.palombi@uniroma1.it (M.P.); sara.trivigno3@gmail.com (S.T.); marcoval.mariani@gmail.com (M.V.M.); paolo.severino@uniroma1.it (P.S.); roberto.badagliacca@uniroma1.it (R.B.); fabio.miraldi@uniroma1.it (F.M.); dario.vizza@uniroma1.it (C.D.V.); 2Department of Nephrology and Dialysis, L. Parodi—Delfino Hospital, 00034 Rome, Italy; dilulloluca69@gmail.com; 3UOC Cardiologia, Ospedale San Eugenio, 00144 Rome, Italy; f.summaria@gmail.com; 4Department of Medicine, Division of Nephrology, Ente Ospedaliero Cantonale, 6900 Lugano, Switzerland

**Keywords:** heart failure with reduced ejection fraction, worsening heart failure, randomized controlled trial, guideline-directed medical therapy, heart failure hospitalization, cardiovascular death

## Abstract

Patients with an established diagnosis of heart failure (HF) with reduced ejection fraction (HFrEF) are prone to experience episodes of worsening symptoms and signs despite continued therapy, termed “worsening heart failure” (WHF). Despite guideline-directed medical therapy, worsening of chronic heart failure accounts for almost 50% of all hospital admissions for HF, and patients experiencing WHF carry a substantially higher risk of death and hospitalization than patients with “stable” HF. New drugs are emerging as arrows in the quiver for clinicians to address the residual risk of HF hospitalization and cardiovascular deaths in patients with WHF. This question-and-answer-based review will discuss the emerging definition of WHF in light of the recent clinical consensus released by the Heart Failure Association (HFA) of the European Society of Cardiology (ESC), the new therapeutic approaches to treat WHF and then move on to their timing and safety concerns (i.e., renal profile).

## 1. Introduction

Heart failure (HF) is a progressive and degenerative disease characterized by a variable duration of symptomatic stability that evolves in episodes of worsening despite continued therapy. These periods are increasingly recognized as a definite phase in the history of HF, termed worsening HF (WHF) [1]. 

Heart failure with reduced ejection fraction (HFrEF) is a major burden for the healthcare system, accounting for almost 50% of incident HF overall [2]. In the last decade, the pharmacotherapy of HFrEF has flourished with new disease-modifying drugs leading to the recognition of four pillars in its management: beta-blocker (bb), angiotensin-converting enzyme inhibitor (ACEi)/angiotensin receptor-neprilysin inhibitor (ARNI), mineralocorticoid receptor antagonist (MRA), and sodium-glucose cotransporter-2 inhibitor (SGLT2i).

Nevertheless, the prognosis of patients with HFrEF treated with guideline-directed medical therapy (GDMT) remains poor, with a high residual risk of cardiovascular death (CVD) or HF hospitalization; tackling this residual risk with novel drugs has become the main objective of randomized controlled trials (RCTs).

The present review aims: (a) to shed light on the new definition of WHF, (b) to introduce the new therapeutic strategies available to treat WHF, (c) their timing, and (d) renal profile.

## 2. Worsening Heart Failure: A Clue to Unravel Clinical Deterioration?

### 2.1. An Evolving Definition

WHF can be defined as exacerbating signs and symptoms of HF in patients with a previous diagnosis of HF needing intensification of treatment, usually diuretic therapy [1]. The definition of WHF has progressively changed from a standpoint that only took into account hospitalizations to now include non-hospitalization events. To date, HF hospitalization and death have been largely used as primary endpoints in most HF clinical trials, and only recently have some studies begun including WHF despite the site of care. Consequently, terms such as acute decompensated HF (ADHF) and acute HF (AHF) have often been considered synonyms for WHF in major HF RCTs, as shown in Table 1 by their inclusion criteria [3,4]. 

The almost mutual use of these terms can be partially explained by the lack of a clear definition of background medical therapy. Considering optimal medical therapy (OMT) (i.e., receiving all available therapies at target doses) as a criterion to potentially qualify the exacerbating signs and symptoms of HF as WHF can be impractical, as a significant number of patients in major HF RCTs and real-life scenarios are treated with “some” background therapy due to intolerance, comorbidities and/or contraindications [1]. 

Recently, a clinical consensus statement released by the Heart Failure Association (HFA) of the European Society of Cardiology (ESC) has provided a comprehensive definition of WHF, in line with current evidence [5]. According to this definition, many efforts are underway to pinpoint patients with WHF in terms of clinical course, setting of care, and subclinical features.

### 2.2. Clinical Course of WHF

The clinical course of HF is characterized by a downward trajectory interspersed by episodes of WHF and acute decompensation requiring escalation of outpatient treatment, emergency department (ED), or in-hospital care [6]. 

Conversely, from WHF, patients with de novo AHF seem to show a distinct clinical phenotype. Generally, the patients are younger, with a lower comorbidity burden, HF being less frequently associated with an ischemic etiology [7]. Patients tend to present with higher baseline blood pressure, better baseline renal function, and functional status, showing superior post-discharge outcomes compared to patients with established WHF [8]. Conversely, mounting evidence derived from data of implantable devices now suggests that ADHF could be a pinnacle of a WHF that has occurred over weeks; pressure changes detected by implantable hemodynamic monitoring (IHM) allow the detection of the transition from chronic compensated to ADHF [9]. In a substudy of the COMPASS-HF trial, Zile et al. found that IHM-derived pressure begins rising 60 days before a hypervolemic HF-related event and continues 14 days after the event. In that way, ADHF may be properly considered a culmination event of WHF resulting from progressive insidious congestion [9].

### 2.3. Setting of Care

At the present time, the outpatient treatment of WHF is generally defined by two primary strategies, including assignment to an outpatient HF care unit for intravenous (IV) diuretic therapy or indication for an augmented oral diuretic or vasodilator regimen [3]. Despite evidence for a greater risk of death associated with WHF following hospitalization, ambulatory care and ED visits for WHF are associated with poor prognosis, significantly worse than HF patients without a WHF episode [1]. 

Notably, not all patients presenting to the ED due to WHF are consequently hospitalized. HF duration is crucial in determining those patients with WHF admitted to ED who will be discharged without hospitalization. Moreover, it should be noted that large variations in hospitalization rates across different regions exist, partly due to nonclinical and non-biological features such as the accessibility to outpatient care facilities or financial deterrent of hospitalizations and family support rather than the actual severity of the disease [1]. 

Thus, Bozkurt et al. have proposed a definition of WHF that is not determined by the acuity or the location of care but by the need for intensified or escalated therapies beyond the standard optimized diuretic therapy [10]. 

While outpatient IV diuretic therapy is broadly recognized as a definite feature in diagnosing and treating episodes of WHF, outpatient oral diuretic treatment has been poorly pinpointed. However, compelling evidence indicates that the necessity of an increased oral diuretic dose in an ambulatory setting is not benign and bears a considerably higher risk of morbidity and mortality [1]. 

### 2.4. Subclinical Features

A limitation of the current WHF definition is that the absence of both signs and symptoms is not always associated with a lower risk. In this regard, Greene et al. have suggested a definition of WHF, which may consider the decline of HF signs or symptoms rather than signs and symptoms [1]. 

Congestive HF can manifest as pulmonary and/or systemic congestion, leading to different pathophysiological implications. Pulmonary congestion is the result of elevated left ventricular filling pressures (LVFP), resulting in symptoms (i.e., dyspnea, orthopnoea, fatigue) and signs (i.e., gallop rhythm on auscultation, pulmonary crepitations, pleural effusion). Systemic congestion refers to an augmented central venous pressure (CVP) due to right-sided HF, leading to peripheral edema (more frequently, ankle swelling), weight gain, jugular venous distention, hepatomegaly, and ascites [11].

Worsening pulmonary congestion can be unrecognized and undertreated, particularly when concealed by the decrease in patient activity to mask the development of manifest symptoms. Likewise, “silent” systemic congestion can be difficult to detect unless it shows itself with clear signs. 

Fluctuations in biomarkers serum levels may help clinicians detect congestion and WHF at an earlier stage so that prompt treatment may prevent adverse outcomes [5]. B-type natriuretic peptide (BNP) or N-terminal pro-B-type natriuretic peptide (NT-proBNP) concentrations have a powerful prognostic role, their changes reflecting the transmural wall stress, a surrogate of disease progression in many “stable” patients [12]. However, NT-proBNP and BNP levels have been studied and validated as diagnostic and prognostic tools to mainly identify and manage patients with HF and left ventricle involvement. Differently, their use in patients with right ventricular heart failure (RVHF) has not been routinely tested. NT-proBNP and BNP elevation cannot be measured to differentiate the relative involvement of each cardiac chamber in the development of overt HF [13]. Currently, no specific biomarker has been validated in order to detect RVHF; carbohydrate antigen 125 (CA 125) could have a potential role in this setting, as it is released by serous tissue in response to fluid overload [14]. In advanced stages of RVHF, indirect signs may be represented by impaired liver function, expressed as albumin serum level reduction and augmented international normalized ratio (INR), the transaminase levels rising significantly in the acute setting of RVHF. A sign of augmented CVP is a rise in serum creatinine, reflecting the worsening of renal function due to elevated renal interstitial pressures and neurohormonal activation [13]. 

Echocardiography could help clinicians in assessing both pulmonary and systemic congestion by measuring inferior vena cava (IVC) diameter and collapsibility and pulmonary artery pressure (PAP), which are representative of ventricular filling pressure and diastolic function such as the E/e′ ratio [5]. In particular, IVC reflects the right atrial pressure, and its augmented size and reduced collapsibility are markers of systemic congestion [15]. 

Recently, the venous excess ultrasound (VExUS) score incorporating IVC size, hepatic, portal, and intrarenal venous Doppler has been adopted in the intensive care unit (ICU) to assess the severity of systemic venous congestion [16]. It is the authors’ opinion that this tool could be implemented in outpatient settings in order to help clinicians manage misleading clinical scenarios.

IHM systems play a pivotal role in detecting pulmonary congestion when still subclinical. A strategy based on PAP measurement in addition to clinical signs and symptoms has proved to ameliorate HF management in patients with New York Heart Association (NYHA) class III [17]. Conversely, a strategy based on impedance-guided management from cardiac resynchronization therapy and implantable cardioverter-defibrillator has shown no reduction in HF hospitalizations [18].

## 3. Worsening Heart Failure: Is It Time to Adopt New Strategies Alongside the “Four Pillars” to Reduce the Residual Risk of Adverse Events?

Several pharmacological treatments have been developed and approved in the last decades for the treatment of HFrEF, leading to a significant reduction in CVD and HF hospitalization [19]. Newly available therapies changed the natural history of this disease, with an increase in the prevalence of HFrEF patients among the general population; as a consequence, physicians face more advanced stages of this challenging disease in their clinical practice. Despite effective new drugs, patient outcomes remain poor and similar to those of some common cancers [20]. 

HFrEF patients have a high residual risk of adverse outcomes (i.e., disease progression, HF hospitalization, and cardiovascular (CV) mortality), even when treated with GDMT with a BB, renin-angiotensin-aldosterone system (RAAS) inhibitor, MRA, and SGLT2i, titrated to the maximum tolerated doses and clinically stable [21,22].

The authors of the 2018 American College of Cardiology/American Heart Association (ACC/AHA) cholesterol guidelines classified the risk of atherosclerotic cardiovascular disease (ASCVD) events (myocardial infarction (MI) or ischemic stroke) in five risk categories, expressed as risk per year. Individuals with previous ASCVD events or a single ASCVD event and high-risk conditions (i.e., diabetes mellitus, hypertension, chronic kidney disease) have a risk of MI or ischemic stroke ≥ 5%. Differently, even the most “stable” HFrEF patients with mild symptoms and no recent HF hospitalizations have a higher risk of CVD and HF hospitalization (≥10% per year) compared to the “very high risk” patients with ASCVD. This residual risk is 14.3% and 12.3% in patients on quadruple therapy with dapagliflozin and empagliflozin as SGLT2i, respectively [23,24]. Furthermore, patients with HFrEF and a recent HF hospitalization or WHF are considered at “very extreme high risk”; the outcome is even worse in patients with advanced HFrEF who are intolerant or refractory to GDMT or have experienced recurrent HF hospitalizations. This comparison is necessary to convey the therapeutic urgency to properly treat patients with HFrEF by applying disease-modifying “quadruple therapy” (BB, ACEi/ARNI, MRA, SGLT2i) [11].

The main target in the treatment of HF is the neurohormonal antagonism; BBs, ACEis, angiotensin receptor blockers (ARBs), and MRAs have been the cornerstones of this therapeutic strategy [25].

Later, in 2014, the PARADIGM-HF trial was conducted [26]; in this RCT, sacubitril/valsartan, an ARNI, was superior to enalapril in the reduction of CVD and hospitalization for HF in patients with chronic HFrEF and left ventricular ejection fraction (LVEF) ≤ 40% (changed to LVEF ≤ 35% during the study). Patients had NYHA class II–IV, augmented values of BNP/NT-proBNP, and an estimated glomerular filtration rate (eGFR) ≥ 30 mL/min/1.73 m^2^. The trial was stopped early after a median follow-up of 27 months due to a lower occurrence of the primary outcome, enhancing the effectiveness of neurohormonal antagonism. 

More recently, new drugs have been developed for the management of HFrEF, paving the way to new pathophysiological targets other than neurohormonal antagonism [27]. 

SGLT2is are novel drugs in the HF scenario, as they were born as anti-diabetic drugs. They work through the inhibition of SGLT2, which mediates about 90% of glucose reabsorption in the proximal renal tubule: SGLT2 blockade induces sodium and glucose loss in the urine, resulting in a natriuretic, diuretic, and antiglycemic effect. In the DAPA-HF trial [28], patients with stable ambulatory chronic HFrEF and LVEF ≤ 40% were randomized to dapagliflozin or placebo; the trial included patients with elevated plasma NT-proBNP levels and an eGFR ≥ 30 mL/min/1.73 m^2^.

On the other hand, the EMPEROR-Reduced trial tested the benefit of empagliflozin vs. placebo in patients with NYHA class II–IV and HFrEF despite OMT [29]. Patients participated in the trial if they had a history of HF hospitalization <12 months; compared to DAPA-HF patients, the study population had a lower LVEF and a higher NT-proBNP level, in addition to a lower cut-off of eGFR (eGFR ≥ 20 mL/min/1.73 m^2^ vs. eGFR ≥ 30 mL/min/1.73 m^2^ in DAPA-HF and PARADIGM-HF). Both dapagliflozin and empagliflozin are now part of the GDMT: the “four pillars” (BBs, ACEis/ARNI, MRAs, and SGLT2is) have proved to reduce the risk of CVD and HF hospitalization in patients with HFrEF with NYHA class II–IV [11].

Omecamtiv mecarbil (OM) and vericiguat are the most recent emerging pharmacologic therapies for HFrEF; as for SGLT2is, they have different mechanisms of action than neurohormonal modulation. Furthermore, their pivotal trials (respectively, GALACTIC-HF [30] and VICTORIA [31]) have been the first RCTs to recruit HF patients with recent hospitalization, i.e., in their “vulnerable phase” [32].

OM is the first direct myosin activator drug targeting impaired myocardial contractility; it differs from classical inotropes because it enhances contractility without altering calcium equilibrium homeostasis or increasing myocardial oxygen demand. The GALACTIC-HF trial [30] randomized patients to OM or placebo; both outpatients and stable inpatients were enrolled, with NYHA class II–IV and LVEF ≤ 35% for ≥ 30 days, elevated plasma NT-proBNP levels and an eGFR ≥ 20 mL/min/1.73 m^2^. The results of this trial were modest; there was no difference in the composite endpoint of CVD and HF hospitalization, but there was a downward trend in HF events; at present, the US Food and Drug Administration (FDA) has rejected to approve OM for treatment of patients with chronic HFrEF.

Another available therapy targeting myocardial contractility without increasing myocardial oxygen demand is the Optimizer Smart device for cardiac contractility modulation (CCM) [33]; this is an innovative intracardiac device-based therapy approved by United States (US) FDA for the treatment of patients with chronic HF, LVEF between 25% and 45%, QRS < 130 ms who remain symptomatic despite OMT. The device releases non-excitatory electrical signals delivered during the cardiac absolute refractory period as an “electrical therapy” that increases contractility, improves cardiac myocyte calcium handling, and modifies gene expression profiles. Different clinical trials demonstrate that CCM could be a device-delivered therapy in managing patients with HFrEF [34,35]. 

Vericiguat is the first medication approved by the European Medicines Agency (EMA) tested in patients with WHF while already on GDMT, with proven efficacy in reducing clinical events [31]. In HFrEF, there is an impairment of nitric oxide (NO)—soluble guanylate cyclase (sGC)—cyclic guanosine monophosphate (cGMP) pathway: oxidative stress, endothelial dysfunction, and inflammation cause a reduction of NO levels, with negative effects on vascular tone, myocardial stiffness, and fibrosis. Vericiguat acts as a direct sGC stimulator. It enhances cGMP independently of NO levels with anti-hypertrophic, anti-fibrotic, and vasodilatory effects [21,36]. 

The VICTORIA trial [31] randomized patients to vericiguat 10 mg vs. placebo. The study included patients with NYHA class II–IV and LVEF < 45%, experiencing an HF hospitalization in the last 6 months or receiving IV diuretics as outpatients in the last 3 months. Exclusion criteria were ADHF, systolic blood pressure < 100 mmHg, eGFR < 15 mL/min/1.73 m^2^, concomitant use of long-acting nitrates or phosphodiesterase 5 inhibitors. The study enrolled a very high-risk population: most patients had NYHA class III–IV, and their NT-proBNP levels were higher than other RCTs [37]. Vericiguat reduced the primary outcome of CVD and HF hospitalization by 10%. The absolute risk reduction (ARR) in the primary outcome was 4.2 events/100 patient/years, with a number needed to treat (NNT) of 24 to prevent 1 composite event over a year; these are remarkable data when compared to the results of other landmark RCTs [Figure 1], even though they enrolled a different risk population. 

Despite the reduction of the primary composite endpoint, vericiguat did not reduce the incidence of CVD alone, while a statistically significant reduction of HF hospitalization was observed in the vericiguat arm. It should be considered that the VICTORIA trial had a shorter duration of follow-up compared to the other trials, as the primary outcome was achieved after a median follow-up of 10.8 months, compared with 27 months for PARADIGM-HF and 18 months for DAPA-HF; this might not have been enough to demonstrate a reduction in CVD [Figure 1]. The VICTORIA trial enrolled a substantial number of patients with WHF who had by themselves poor prognosis. The data show that “relatively stable” patients with a lower baseline NT-proBNP might benefit most from vericiguat administration [25,32].

Lower-risk HFrEF patients will be studied in the ongoing phase III VICTOR trial. Given the very positive results coming from the VICTORIA trial, the 2021 ESC Guidelines for the diagnosis and treatment of acute and chronic HF recommend the use of vericiguat for patients with recent episodes of WHF during GDMT (Class IIb, Level of Evidence B) [11]. 

## 4. The Damaging Course of Heart Failure: Can We Slow the “Rolling Stone”?

Patients with HF have to struggle against chronic disease, alternating between steady-state and hospital readmissions for acute episodes; every exacerbation of the disease leads to potentially irreversible loss of capital, in a one-way road to end-stage HF and fatal outcome [32]. Time seems to be crucial in decreasing this damaging course and improving prognosis: Abdin et al. coined the expression “time is prognosis” to underline the urgency to treat HF promptly with outcome-modifying therapy, i.e., GDMT [38].

AHF is a life-threatening condition associated with an in-hospital mortality of 4–6%; these mortality rates are even higher in the immediate period after discharge, referred to as the “vulnerable phase” (10–30% 1-year mortality and readmission of one-third of patients within 6 months post-discharge), whereas chronic outpatients have a 40–50% 5-year mortality rate [38,39]. The interval between pre-discharge and post-hospitalization is critical and requires prompt recognition and treatment of congestion to avoid premature readmission; for this purpose, results from two major RCTs suggest that the addition of acetazolamide [40] or hydrochlorothiazide [41] to IV loop diuretics in patients with AHF is associated with a greater incidence of successful decongestion during hospitalization. Simultaneously, an optimization of GDMT is needed to improve long-term prognosis [15,39]. 

However, only a few drugs have been tested in patients in the vulnerable phase after hospitalization for HF; in fact, RCTs are usually performed in chronic stable HF outpatients with at least a 3-months-unchanged therapy as inclusion criteria [15,38,42,43,44,45,46,47,48,49,50,51,52,53]. On the other hand, many clinical trials performed in the pre-discharge and post-hospitalization phase showed neutral results and did not reach their primary endpoints [54,55]. 

Among these RCTs [Table 1], the PIONEER-HF trial recruited HFrEF patients during hospitalization for ADHF; they were randomized to sacubitril/valsartan or enalapril after reaching hemodynamic stability. In this trial, sacubitril/valsartan resulted in greater cardiac unloading, as suggested by a larger reduction in NT-proBNP levels compared to enalapril and a reduction of HF rehospitalizations, CVD and heart transplantation; it was not powered for clinical endpoints but still signals for risk reduction are shown [56].

Several trials studied the benefits of the early introduction of SGLT2is in the medical therapy of patients hospitalized for AHF. In the EMPULSE trial, empagliflozin 10 mg die versus placebo was started after initial stabilization (median time of 3 days after admission) in patients with ADHF, regardless of their LVEF or diabetes status. The primary outcome was a composite of all-cause death, number of HF events and time to first HF event, and a ≥5 change from baseline in the Kansas City Cardiomyopathy Questionnaire total symptom score. The patients treated with empagliflozin had clinical benefit compared with placebo at 90-day follow-up [57]. Similarly, the SOLOIST-WHF trial enrolled patients with type 2 diabetes mellitus who were recently hospitalized for WHF. They were randomized to sotagliflozin or placebo, initiated before or shortly after discharge; sotagliflozin therapy resulted in significantly fewer CVD, hospitalizations, and urgent visits for HF compared with placebo [58].

The 2021 ESC Guidelines for the diagnosis and treatment of acute and chronic HF emphasize the importance of comorbidities treatment in HF patients [11]. In the AFFIRM-AHF trial, patients hospitalized for HF with LVEF < 50% and iron deficiency were randomized to IV ferric carboxymaltose or placebo. Administration of ferric carboxymaltose did not significantly reduce the primary composite outcome of total HF hospitalizations and CV death at 52 weeks. However, the trial showed that in HFrEF patients with iron deficiency, after stabilization post-acute heart failure, IV ferric carboxymaltose was safe and reduced the total number of HF hospitalizations compared with placebo [59].

To date, the VICTORIA trial is the only large positive RCT designed to specifically enroll patients with worsening HFrEF in the vulnerable phase immediately post-discharge [31]. In the secondary analysis of the VICTORIA trial, Lam et al. showed that there is a gradient in the risk of clinical events in HF patients that is higher at the time of discharge and within the first 3 months after hospitalization and decreases exponentially over time. Patients recruited within 3 months after hospitalization had twice the incidence of the primary endpoint compared to those recruited within 3–6 months; however, vericiguat was equally effective in vulnerable HF patients regardless of the time of hospitalization [60,61]. 

Evidence supports the introduction of vericiguat immediately before discharge, as the VICTORIA criteria may be applicable in 40% of patients admitted for AHF [15,39]. 

Recently, Greene et al. [62] proposed a practical clinical approach to properly treat worsening HFrEF: ARNI, BB, MRA, and SGLT2i are the “4 pillars” of this strategy. 

There is compelling evidence supporting rapid sequence or simultaneous initiation of quadruple medical therapy following a WHF event, both in hospitalized patients who have been stabilized and outpatients. The STRONG-HF trial enrolled patients admitted to hospital with AHF, not treated with full doses of GDMT [63]; this is the first RCT providing direct evidence of the efficacy, safety, and pharmacological tolerability of rapid sequence or simultaneous initiation and titration of GDMTs in patients hospitalized for HF. The trial was stopped earlier because of the indisputable efficacy in the reduction of 180-day death or HF hospitalization. This trial highlights the benefit of early combination therapy; subsequently, the delay in the initiation of these four medications exposes eligible patients to augmented clinical risk without reason. Clinicians should prioritize the initiation of low doses of each class of medications over the escalation of any one of them; as a second step, all eligible GDMTs should be titrated to their targets within 4–6 weeks of the WHF event, prioritizing escalation of BB as tolerated.

Balestrieri et al. [64] revised the definition of WHF with the inclusion of optimal background therapy as a rule to consider an exacerbation of signs and symptoms as WHF in order to homogenize groups of patients in RCTs [Table 1]. Given the residual risk of adverse CV events in patients with WHF despite quadruple therapy, the introduction of vericiguat rises as the “fifth card” to play in patients with WHF due to its different mechanism of action, i.e., its additive benefit, its safety and tolerability (“quintuple therapy”). 

However, the benefits of vericiguat on clinical outcomes are robust regardless of the background use and dose of GDMT [62]. Early initiation of vericiguat should be considered in combination with rapid sequence or simultaneous optimization of GDMT with the four pillars as tolerated. Alternatively, clinicians could optimize quadruple medical therapy after a WHF event and use vericiguat for a following WHF event. Moreover, vericiguat could be considered for early use in patients with WHF and contraindications or intolerance to drugs of quadruple therapy [62].

## 5. Worsening Renal Function: How Far Can We Go with Optimal Medical Therapy?

Chronic kidney disease (CKD) accounts for one of the most important comorbidities in patients with CV disease, with worsening renal function being a crucial limiting factor in achieving optimal medical therapy in HF patients. 

The importance of prompt treatment of comorbidities in HF is underlined in the 2021 ESC Guidelines for the diagnosis and treatment of acute and chronic HF and in the 2023 Focused update [11,65]. Finerenone, a new non-steroidal MRA, was recommended to reduce the risk of HF hospitalization in patients with CKD associated with type 2 diabetes without a history of chronic HF. Data supporting this recommendation come from FIDELIO-DKD [66] and FIGARO-DKD trials [67], which confirmed the reduction in composite CV outcome associated with finerenone, mainly driven by reductions in HF hospitalizations. These studies underline the cardioprotective effect of finerenone in CKD patients with type 2 diabetes mellitus, a population at high risk of CV disease development. They are thought-provoking regarding the possible role of finerenone in the prevention of the development of symptomatic HF [68].

Patients with an established diagnosis of HF are at risk of deterioration of renal function as a consequence of cardiac and/or renal disease progression [5]. Worsening of renal function is frequently accompanied by hyperkalemia [69], a condition which leads clinicians to down titrate or even withdraw ACEi, ARBs, and MRAs, promoting a decline in cardiac function with a subsequent increased risk of CVD and hospitalization for HF [70].

In this damaging loop, vericiguat could act as a safe drug in patients with severe impairment of renal function (eGFR ≥ 15 mL/min) [31] [Figure 2].

In the development of CKD, cGMP deficiency seems to represent one of the pathophysiological mechanisms accountable for the progression of renal disease [31]: direct stimulation of sGC could be a crucial therapeutic target for the management of CKD due to vasodilation of glomerular arterioles and consequent reduction in the degree of endothelial dysfunction [31].

A recent article by Voors et al. evaluated the relationship between vericiguat efficacy and changes in renal function in patients enrolled in the VICTORIA trial [71]. The data showed that in patients with severe HF with a marked reduction in LVEF and very high CV risk, the curves for trends in renal function parameters were similar in the vericiguat and placebo groups. Therefore, the primary composite endpoint (CV death and hospitalization for HF) was reduced in the vericiguat group across a wide range of eGFR (from 15 to 60 mL/min). The beneficial effects of vericiguat appear to be the same both in patients with substantially preserved renal function (eGFR > 60 mL/min) and in those with later stages of renal disease (eGFR < 30 mL/min) [71].

Additionally, treatment with vericiguat does not impact serum potassium levels, and it can be administered even to patients with hyperkalemia for whom RAAS inhibitor therapy is contraindicated [71].

The results of the VICTORIA trial, in terms of renal survival, show how vericiguat does not impact renal function curves, even in the most advanced stages, being able to be administered even in patients with eGFR < 30 mL/min. At present time, there are no data on patients with end-stage CKD (eGFR < 15 mL/min) and those on renal replacement treatments [31]. In the difficult context of patients with refractory (i.e., therapy-resistant HF), ultrafiltration could play a crucial role in managing fluid balance. Although the literature sometimes presents seemingly controversial data, there is no doubt that dialysis removal of water and electrolytes contributes to symptom relief. The UNLOAD [72] and AVOID—HF [73] clinical trials clearly demonstrated how the use of ultrafiltration resulted in better management of fluids by reducing body weight and improving renal outcomes.

In conclusion, maximal and prompt treatment of WHF patients is a major challenge, or probably the only possible to date, as natural WHF progression results in frequent rehospitalizations and symptoms’ exacerbation up to the advanced stage, in which commonly available therapies are no longer effective or sufficient [74,75]. Positive trials on this high-risk population with advanced HF are lacking, and pharmacological options are poor [30,75,76,77]. Time plays an important role in WHF prognosis, and the introduction of new effective drugs should not be postponed.

## Figures and Tables

**Figure 1 jcm-12-06956-f001:**
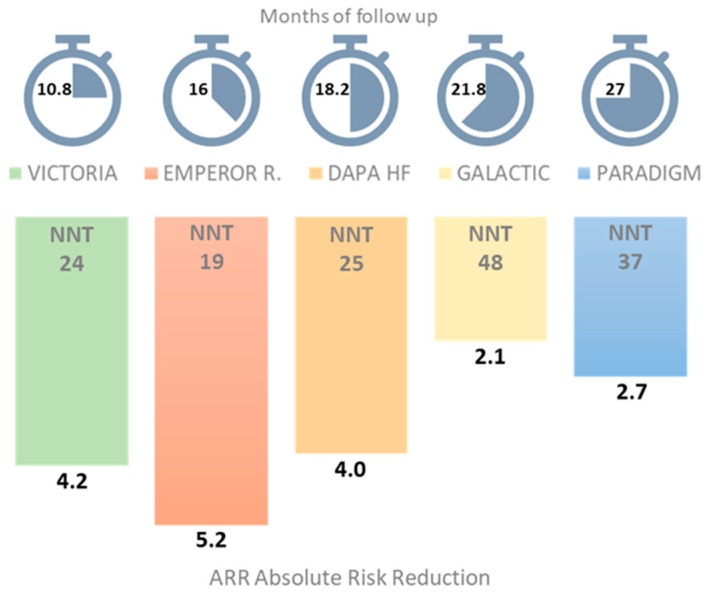
Months of follow-up for each study related to their composite primary endpoint: heart failure hospitalization (HFH) or cardiovascular death (CVD). NNT, number needed to treat.

**Figure 2 jcm-12-06956-f002:**
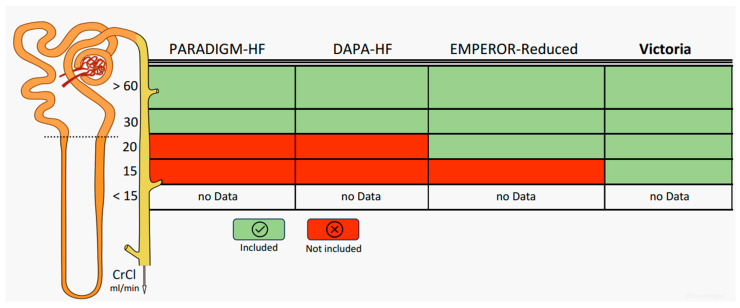
Inclusion and exclusion criteria of patients enrolled in the four registrative heart failure (HF) trials according to creatinine clearance (CrCl). The dashed line represents a CrCl of 30 mL/min.

**Table 1 jcm-12-06956-t001:** Recent HF clinical trials and their inclusion criteria.

Clinical Trial	Drug	Inclusion Criteria
PIONEER-HF (881 pts)	sacubitril/valsartan vs. enalapril	Currently hospitalized for a primary diagnosis of HF, including symptoms and signs of fluid overload; randomized no earlier than 24 h and up to 10 d after initial presentation while still hospitalized; stable as defined by an SBP > 100 mm Hg for the preceding 6 h in the absence of symptomatic hypotension, no increase (i.e., intensification) in IV diuretics or use of IV vasodilators within the last 6 h, and no IV inotropes for 24 h prior to randomization
AFFIRM-HF (1110 pts)	ferric carboxymaltose vs. placebo	Hospitalized with clinical signs, symptoms, and biomarkers consistent with AHF. During the index hospitalization, patients had to have received at least 40 mg of IV furosemide
VICTORIA (5050 pts)	vericiguat vs. placebo	Evidence of WHF (hospitalized within 6 months before randomization) or receiving intravenous diuretic therapy, without hospitalization, within the previous 3 months
GALACTIC-HF (8256 pts)	omecamtiv mecarbil vs. placebo	Currently hospitalized for HF (inpatients) or had either made an urgent visit to the emergency department or been hospitalized for heart failure within 1 year before screening (outpatients). 18 < age < 85
SOLOIST-WHF (1222 TDM2 pts)	sotaglifozin vs. placebo	Hospitalized because of the presence of signs and symptoms of HF and received treatment with intravenous diuretic therapy. 18 < age < 85
EMPULSE (530 pts)	empaglifozin vs. placebo	Admitted to the hospital for AHF after initial stabilization (SBP ≥ 100 mmHg and no symptoms of hypotension in the preceding 6 h, no increase in i.v. diuretic dose for 6 h prior to randomization, no i.v. vasodilators including nitrates within the last 6 h prior to randomization, no i.v. inotropic drugs for 24 h prior to randomization)
ADVOR (519 pts)	iv acetazolamide vs. placebo	Hospitalized for acute decompensed HF with clinical signs of fluid overload treated with iv loop diuretics (iv dose twice the oral manteinance dose)
CLOROTIC (230 pts)	hydrochlorothiazide vs. placebo	Hospitalized ≤ 24 h for acute decompensed HF, treatment with an oral loop diuretic ≥ 1 months before hospitalization

HF, heart failure; AHF, acute heart failure; WHF, worsening heart failure; SBP, systolic blood pressure; IV, intravenous; PTS, patients.

## Data Availability

Not applicable.

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
