# Peer review of "New Challenges in Heart Failure with Reduced Ejection Fraction: Managing Worsening Events"

_jcm, 2023, doi:10.3390/jcm12226956_

Round 1

Reviewer 1 Report

Comments and Suggestions for Authors

Managing deteriorating events in heart failure with decreased ejection fraction is the main emphasis of this comprehensive review, which also addresses other crucial issues. However, some specific critiques deserve attention:

-       I find some content similarities with the recently published clinical consensus statement on the same topic by the Heart Failure Association of the European Society of Cardiology (Eur J Heart Fail 2023;25:776–791): what is the added value of the present review compared to this document?

-       In my opinion, the text should be significantly shortened; in particular, being a question-and-answer-based review, it is not necessary to re-describe the clinical studies in depth, while I find it more innovative to focus on the management strategies;

-       As always, nonembolic acute right ventricular failure is somewhat "forgotten": I propose to differentiate pulmonary congestion from systemic congestion, which should be investigated in the same way as a cause of worsening heart failure, considering that right ventricular dysfunction is a field of recent renewed interest, but its prevention, treatment, and, in part, natural history remain largely unexplored; 

-       Venous congestion commonly occurs in acute right ventricular failure, but accurately quantifying systemic congestion is difficult and may commonly be missed unless it is obvious. There are currently no biomarkers specific for right ventricular failure. Therefore, it would be interesting to know the author's opinion regarding the potential contribution of the portal vein and intrarenal vein echocardiography in the management of patients with worsening heart failure.

Reviewer 2 Report

Comments and Suggestions for Authors

1. This manuscript's figures and tables are directly taken from the references. Did authors get permissions from the original manuscript authors?

 2. WHF is well defined in manuscript reference 1. Is there really a need to go over the same if this manuscript is supposed to be a treatment overview?

3. JACC has published a comprehensive review on the same topic. DO we need another review on this topic?

Comments on the Quality of English Language

Minor spell check 

Reviewer 3 Report

Comments and Suggestions for Authors

This is an interesting article that provides an overview of data from RCTs in HFrEF. There are 3 aspects that needed to be captured in this work.

1.                     The role of combined diuretic therapy and some recent RCTs that investigated the potential benefits of acetazolamide and hydrochlorothiazide in acute decompensated HF.

2.                     The emerging role of novel non-steroidal MRAs in HF. In the 2023 update of the ESC guidelines, the non-steroidal MRA finerenone has received a recommendation for use in patients with T2M and CKD with the aim to reduce the risk of HF hospitalization.

3.                     Which is the role of isolated ultrafiltration in the management of worsening HF?

Round 2

Reviewer 1 Report

Comments and Suggestions for Authors

The authors have responded adequately to the previously reported criticisms. The paper can be accepted in this version.